# Climatic conditions are weak predictors of asylum migration

Sebastian Schutte [1,3], Jonas Vestby[1,3], Jørgen Carling [1] & Halvard Buhaug [1,2,3 ✉]

Recent research suggests that climate variability and change significantly affect forced migration, within and across borders. Yet, migration is also informed by a range of non-climatic factors, and current assessments are impeded by a poor understanding of the relative importance of these determinants. Here, we evaluate the eligibility of climatic conditions relative to economic, political, and contextual factors for predicting bilateral asylum migration to the European Union—a form of forced migration that has been causally linked to climate variability. Results from a machine-learning prediction framework reveal that drought and temperature anomalies are weak predictors of asylum migration, challenging simplistic notions of climate-driven refugee flows. Instead, core contextual characteristics shape latent migration potential whereas political violence and repression are the most powerful predictors of time-varying migration flows. Future asylum migration flows are likely to respond much more to political changes in vulnerable societies than to climate change.

[1] Peace Research Institute Oslo, Oslo, Norway. [2] Department of Sociology and Political Science, Norwegian University of Science and Technology, Trondheim, Norway. [3] These authors contributed equally: Sebastian Schutte, Jonas Vestby, Halvard Buhaug. ✉email: halvard@prio.org

Climate variability and extreme weather events affect human migration, directly through the destruction of homes and infrastructure, indirectly through disruption of economic activity and well-being, and preemptively as adaptation to emerging stressors[1–9]. Migratory responses can take on many forms, including temporary displacement, accelerating urbanization, and cross-border migration. Natural hazards may also trap vulnerable populations as their capacity to move erodes. Quantifying the number of people who migrate at least partly due to climate- and weather-related events is inherently challenging, due to conceptual ambiguity, lack of robust methodologies, and poor data[5,9]. In general, there is more evidence that climatic conditions influence short-distance temporary displacement than international migration[8,10]. However, several recent studies have linked adverse climatic conditions in sending countries to refugee and asylum migration, for which statistics are more complete[11–13]. Extrapolating their estimated temperature effect into the long-term future, one study concludes that the volume of asylum applications to the European Union (EU) could triple within this century under a high warming scenario, all else constant[13].

The real-world relevance of such extrapolations can be questioned on two fronts. First, statistically significant correlations derived from in-sample regression models often constitute poor predictors for new data[14–16]. Thus far, we lack systematic evidence of how well climatic factors predict future flows in forced migration under realistic conditions. Second, migration is not monocausal but instead a result of complex interactions between a range of economic, social, political, demographic, and environmental factors[1–3,17,18]. In this context, climatic conditions are generally considered secondary to socioeconomic and political drivers[2,12,19]. Regression-based projections that hold all non-climatic drivers constant fail to reveal the comparative magnitude of the climate effect and can say little about the expected overall volume of migration flows in the future.

Recognizing the limited ability of earlier research in accounting for these complexities, leading scholars of climate change and migration have called for more critical and nuanced approaches, to move beyond simplistic narratives of "climate refugees" and avoid a securitization of the issue[19–21]. One important element in a more careful approach is placing the climate effect in context. To this end, we conduct a quantitative investigation into the relative performance of climatic, economic, political, and contextual factors in predicting near-future asylum migration to the EU.

Mirroring the global trend in forced migration[22], the EU has experienced a dramatic surge in the arrival of asylum seekers in recent years (Fig. 1a), with Afghanistan and the Middle East being major sending areas (Fig. 1b). This recent spike has generated grave humanitarian challenges, with numerous migrants and refugees dying in their attempts to reach Europe. The migrant crisis also has had profound impacts on the European political landscape; societal impacts that have been attributed in part to the growth in asylum immigration include a rise in populism, Euroscepticism, and support for the Brexit movement[23–26].

Although asylum migration traditionally is seen as driven by violent conflict and repression[27,28], climatic stressors might conceivably inform threat perceptions underpinning a decision to flee. Due to high risks and costs associated with displacement[21], citizens in conflict-affected countries not directly threatened by the fighting typically seek to stay and wait out the conflict. However, this decision is only viable as long as the physical environment can sustain their mode of living. When a severe drought strikes a war-torn society, the outcome not only is potentially devastating for local livelihood, economic activity, and food security but also likely to result in increasing displacement and forced migration[29], accentuating the need for international protection. Climate hazards can also generate forced migration indirectly by triggering or escalating conflict and insecurity[11,30], although the prominence of this pathway remains disputed[31–34].

Perhaps most importantly, seeking asylum is the main opportunity for spontaneous migration to high-income countries. Whenever extreme climate and weather events undermine agrarian livelihoods, cause widespread material destruction, or overstrain government systems, responses in the form of international migration could be channeled through the asylum system for lack of alternatives. Over the past decade, the EU has received asylum seekers from every single country in Africa, the Middle East, and Asia, and the majority of these claims have been rejected. In other words, asylum migration is, despite its more narrowly conceived purpose, plausibly responsive to climate-related stressors[13]. Recognizing weather-related threats to basic human rights including right to life, the UN High Commissioner for Refugees (UNHCR) has argued that adverse effects of climate change and natural disasters may produce valid claims for refugee status[35], and the European Commission is currently exploring a potential European framework for "climate refugees"[36].

## Results

**Prediction framework.** Scientific investigations into drivers of forced migration are confronted with a number of non-trivial challenges[5,19,21,37]. Conceptually, motivations to move are complex, dynamic, and endogenous, and they often negate a stylized dichotomy of forced vs. voluntary migration. Analytically, studies must overcome challenges related to omitted variable bias, endogeneity, nonlinear functional forms, unknown statistical interactions, and heterogenous treatment effects.

A common approach to studying impacts of climatic conditions on migration is reduced-form regression analysis[13,38–40]. Such models can be useful if one seeks to estimate the causal effect of a variable of interest, although accurate inference depends on correct specification of functional forms, interactions, and conditions that might shape the effect on the outcome. In the context of asylum migration, accounting for all plausible indirect and interactive processes through which climate might play a role is extremely challenging, suggesting that variable importance and model performance should not be assessed via metrics of in-sample fit alone. Fixed effects regression models are poorly suited to isolate, quantify, and rank endogenous explanatory factors, and such models also do not lend themselves to high-quality prediction out of sample, as unit- and time-fixed effects are naturally subject to change over time. Finding relevant predictors that generalize beyond the statistical sample is difficult in this framework.

To overcome these challenges and facilitate an analysis of the relative capability of climatic conditions in predicting asylum migration, we make use of two technologies from machine learning: random forests (RFs) and leave-future-out cross-validation (LFO-CV). Out-of-sample prediction such as LFO-CV is a powerful heuristic to assess the relative performance of predictors, especially when competing theories and empirical measures are highly collinear. Importantly, the RF model accounts for nonlinear functional forms and complex interactions[16,41,42]. The explicit consideration of multivariate interactions between predictors is an essential property of the RF for our purpose, given theorized connections between climatic, economic, and political factors, and because climatic drivers are believed to be highly context dependent[8]. See "Methods" section for additional details.

The prediction framework cannot be used to estimate causal effects, but it can serve to test observable implications of relevant theories and evaluate the quality and veracity of empirical models. Predictors that perform well on new data are likely to capture

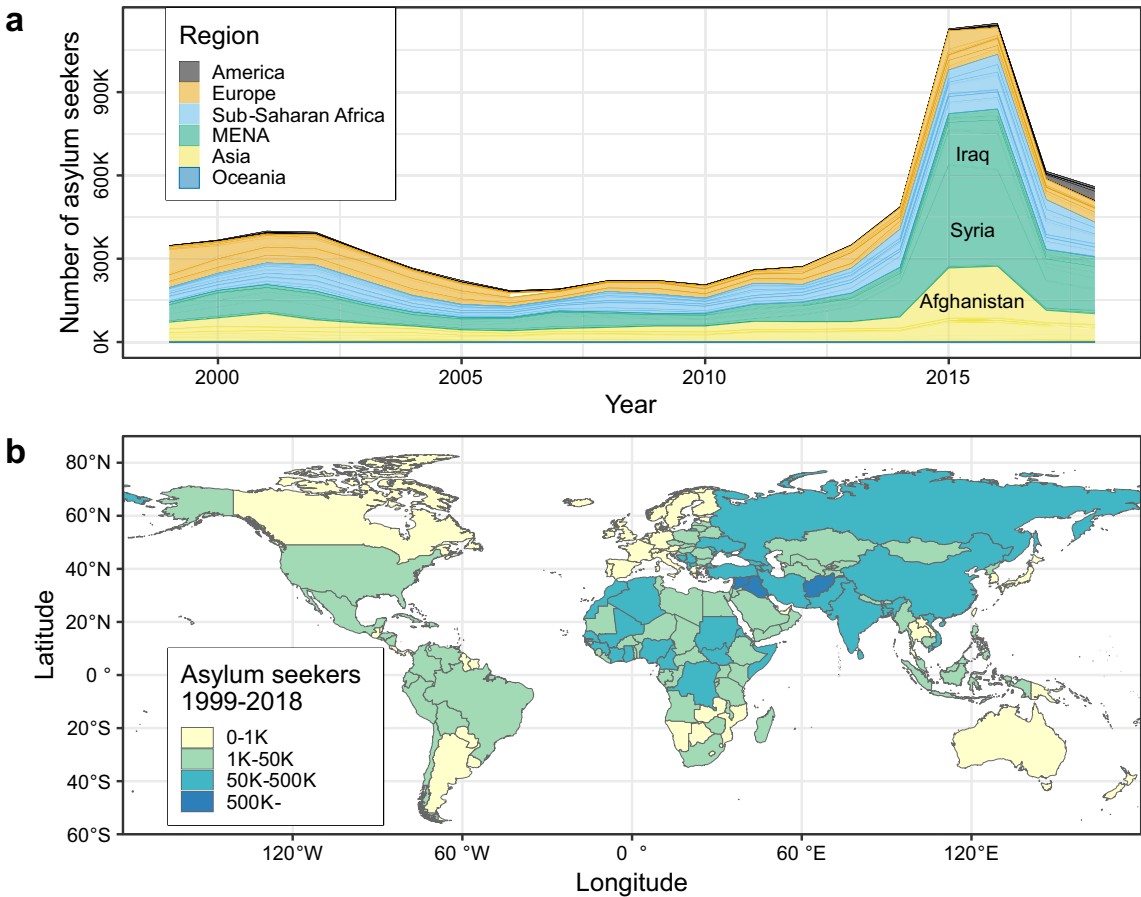

**Fig. 1 Spatiotemporal patterns of asylum migration to the EU. a** Trend in annual number of new asylum applications by country and region of origin, 1999–2018. MENA is the Middle East and North Africa. **b** Total volume of first-time asylum applications submitted to the EU by country of origin, 1999–2018.

important data-generating processes underlying the theoretically informed causal variables, whereas estimated causal effects that fail to predict similar outcomes out of sample might reflect overfitting or misspecification of the original model, or causal processes that are unique to the study sample[16]. Detecting reliable, leading indicators of asylum movements to the EU also carries real-world relevance for decision-makers and further enables more realistic assessments of future migratory responses to different climate-change scenarios and socioeconomic development pathways, e.g., within the scope of climate-change impact assessments.

**Data structure and predictors**. We construct an empirical dataset with yearly observations of 175 independent countries worldwide for all valid years (1999–2018). As the outcome variable, we measure the total number of first-time asylum applications submitted to any EU member state (EU-28) from each country of origin in each calendar year. The variable is adjusted for population size in country of origin. In Supplementary Information Section 3.9, we document results from a global analysis using stock data of asylum applications.

To evaluate the importance of climatic conditions in predicting asylum migration, relative to economic and violence-related factors, we classify three substantive components that represent complementary dimensions of theorized drivers[12,18,28,43,44]. The emphasis here is on time-varying conditions in countries of origin that best predict the onset of forced migration. The climate component captures weather-related impacts on rural and urban

economic activities, as theoretical models of climate-driven migration give prominence to economic mechanisms[5,40]. Effects on agricultural productivity are proxied via positive and negative deviations in the Standardized Precipitation-Evapotranspiration Index (SPEI). We use the 3-month version (SPEI-3), measured exclusively for cropland areas during the local growing-season months, to ensure that the predictors reflect climatic conditions shortly prior to and including the current growing season, and ignore weather conditions during other parts of the year. Severe growing-season drought can cripple rural livelihood and income, especially in rain-fed agricultural systems, whereas above-average rainfall in these regions typically results in a generous harvest[45]. To account for recurring anomalies, we also include positive and negative deviations in 3-year moving average SPEI-3 scores. Further, to capture weather-related impacts on non-agricultural economic activities, we include annual mean temperature. The temperature predictor is weighted by local population density before aggregation to the country level, to represent the sensitivity of urban economies (e.g., worker productivity). Anomalous temperatures and severe drought have been linked to a range of negative human security outcomes[1] and the variables included here capture both short-term shocks and multi-year anomalies.

The economy component measures macro-economic performance and human development, represented by the following five predictors: Gross Domestic Product (GDP) per capita, inter-annual growth in GDP per capita, interpersonal globalization index, share of the young adult population with post-secondary education, and infant mortality rate. Economic wealth and well-being are expected to reduce the need for international

protection, whereas education and connectedness facilitate increased mobility. Although average income levels correlate strongly with average education levels and life expectancy, multicollinearity does not introduce bias in the LFO-CV prediction framework. In addition to theorized direct effects, low development and poor economic prospects also may moderate a climate effect on asylum migration. For example, some earlier research suggests a non-monotonic effect of temperature on migration, with higher migration rates in middle-income countries than in poorer economies[38].

The violence component measures society-level political violence, repression, and crime, and consists of the following five predictors: battle-related deaths, total population size residing within 20 km of armed conflict events, physical integrity rights, freedom of movement, and annual homicide rate. The political and security contexts represented by these indicators not only proxy a well-founded fear of being persecuted, the key element in the legal definition of a refugee, but also may serve as crucial scope conditions shaping the influence of climate-related hazards on asylum migration. Both policy[29] and academic[8] studies highlight how prevailing conflict and insecurity increase forced displacement in response to weather shocks.

To supplement the theoretically informed predictors, we identify a set of baseline indicators, included in all component models, which represent structural and contextual opportunities for migrating to Europe: highest neighborhood democracy score, country area, country population size, extent of urbanization, and geodesic distance to the nearest EU-28 country. See "Methods" section for further details and Supplementary Information Section 2 for descriptive statistics.

In addition to the thematic component models, which consist of the respective component predictors and the baseline variables, we test a full ("all") model comprising all indicators. All predictors are lagged 1 year to ensure proper sequencing of events and to allow for a delay between experienced climatic, economic, and violence-related hazards and the submission of asylum application in Europe. Traveling to Europe through common migration routes often takes many months[46]. In the Supplementary Information, we document results using contemporaneous and 2-year lagged predictors. Regional and temporal trends in these factors are visualized in Fig. 2.

**Relative performance of thematic component models.** As a preliminary assessment of the data and as a direct comparison with recent findings from the empirical literature, we inspect the in-sample association between the component indicators and asylum migration through conventional two-way fixed effects regression (Supplementary Table 2). In line with earlier research, we find a statistically significant U-shaped association between temperature and asylum applications. All economy- and violence-related indicators also reveal significant associations with the outcome. However, because of known dependencies between these indicators, the regression coefficients cannot inform us about the overall importance of, e.g., temperature in driving asylum migration to the EU, relative to other drivers, and concerns about overfitting mean that these results may not translate into good early warning predictors of future forced migration flows. Hence, we rely on out-of-sample prediction as our primary scientific tool.

We assess the relative importance of the thematic component models and the full model by comparing their LFO-CV prediction errors. All models are trained on sliding 4-year windows of data within the range 1999–2017 and tested against observed outcomes for the subsequent year 2003–2018. By covering short periodic subsets of the data in each model run,

repeated for the full temporal domain, the analysis explicitly evaluates the extent to which predictors are robust to small changes in the period of analysis. To account for missing values in the predictors, we rely on multiple imputation procedures, entailing that each model is estimated ten times on slightly different data for each training period.

Figure 3a displays the models' mean absolute error across all simulations. According to this metric, the violence model produces the most accurate out-of-sample predictions, ahead of the economic component model. In fact, the violence model outperforms even the full model composed of all three components and the baseline indicators. The climate model performs the worst in this test, implying that the temperature and SPEI predictors collectively miss important early warning signals that are captured in the violence and economy models.

In Fig. 3b, we plot individual RF model predictions against actual outcomes for the full range of reported asylum applications. For low levels of migration, the competing predictions overlap and align well with observed values. Country-years that produce larger volumes of asylum migration are harder to predict accurately and here the difference in performance between the models is more apparent. Even so, the average predictions follow the same general shape. The modest spread in predictions is determined partly by the influential baseline indicators (notably, distance to EU-28), which are common to all models, and partly by the use of correlated indicators across components that may act as substitutes for each other (e.g., physical integrity rights, which is part of the violence model, correlates negatively with the economy component model's infant mortality rates).

To better assess the relative merit of the components in predicting observed flows in asylum migration, Fig. 3c visualizes the competing RF predictions over time. For each source country in each year, we identify the thematic component model that produces the most accurate prediction and allocate all asylum seekers from that country-year to that model. Summing up by model and calendar year, we find that the sharp increase in asylum migration to the EU since 2012 is best captured by the violence model. The observations represented by the economy and climate models also display a growth in migration levels in recent years, but much less so than the violence model. Likewise, the steep decline in applications since 2016 corresponds with a substantial reduction in conflict severity in major sending areas. The Uppsala Conflict Data Program's best estimate indicates that the global number of people killed in armed conflict events dropped from around 99,000 in 2015 to <54,000 in 2018[47]. Climatic and economic conditions in countries of origin add predictive power primarily in non-conflict settings where the level of asylum migration tends to be low.

**Relative performance of component predictors.** Beyond quantifying aggregate component model performance, we assess the behavior of individual predictors by means of estimating accumulated local effects (ALEs)[48]. ALE functional forms that deviate markedly from a horizontal line denote larger influence on the prediction, although effects should be interpreted locally (i.e., over small windows), as none of the effects is independent. For each imputed dataset and training period, we generate one ALE line, such that the total spread of ALEs reflects degree of prediction consistency across the cross-validation set, i.e., how robust variable associations are over time (Fig. 4).

Several factors that remain constant or change only slowly within individual time-series display steep ALE curves, as they capture systematic cross-country differences in the latent production of asylum seekers. For example, the predicted per-capita level of asylum migration to the EU drops sharply as a

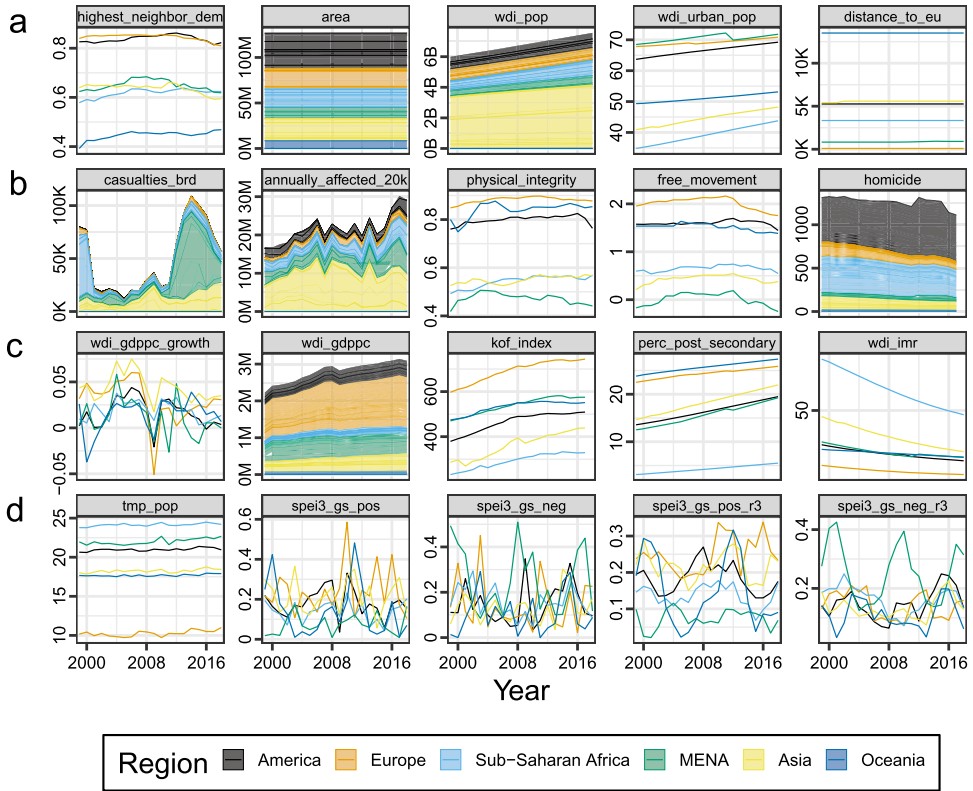

**Fig. 2 Spatiotemporal patterns in drivers of asylum migration, aggregated by region. a** Baseline component: highest democracy score in neighborhood (average score); country area (cumulative km$^2$); country population (cumulative); urbanization rate (average %); distance to nearest EU member state (average km). **b** Violence component: battle-related deaths (cumulative); population size within 20 km of nearest conflict event (cumulative); physical integrity index (average score); freedom of movement (average score); number of homicides (cumulative). **c** Economy component: annual growth in GDP per capita (average rate); GDP per capita (cumulative Int$); interpersonal globalization index (average score); share of population with post-secondary education (average %); infant mortality rate (average). **d** Climate component: population-weighted mean temperature (average °C); positive deviation in growing-season-weighted SPEI-3 index (average $\sigma$); negative deviation in growing-season-weighted SPEI-3 index (average $\sigma$); positive deviation in 3-year moving average SPEI-3 (average $\sigma$); negative deviation in 3-year moving average SPEI-3 (average $\sigma$). MENA is the Middle East and North Africa. $N = 3413$ country-year observations.

function of the distance from Europe, whereas it peaks at a source-country population size of around three million, all else constant (Fig. 4a). From a policy perspective, such insights are of secondary interest, as they refer to inert demographic and geographic features characterizing areas of concern that cannot provide early warning signals of emerging migrant crises. Substantively, these results are testimony to the significant challenges involved in traversing long distances to apply for asylum in Europe, potentially involving the crossing of high mountain ranges (e.g., Afghanistan), inhospitable deserts (e.g., Somalia), hostile neighbors (e.g., Yemen), or dangerous waters (North Africa). In larger countries, internal displacement may be a more viable alternative to costly and risky transnational asylum migration.

Most violence indicators capture conditions that can shift quickly and most reveal important associations with the production of asylum seekers that are robust across time periods (Fig. 4b). Incremental changes from peace to low-intensity fighting contribute little to the prediction, but the estimated number of asylum applications rises log-linearly from around 500 annual battle deaths. This threshold is comparable to the severity level of, e.g., the conflict between the government of Mali and various Islamist groups in northern parts of the country and fighting between the government of South Sudan and forces loyal to former vice-president Machar in recent years[47]. Deterioration in basic civil rights is also strongly predictive of levels of asylum migration, whereas homicide levels matter relatively less. These

results are not an artifact of the violent years following the 2011 Arab Spring uprisings; Supplementary Information Section 3.7 reveals that the violence indicators are superior predictors also when the models are trained exclusively on the pre-2011 period.

Among the economy indicators (Fig. 4c), GDP per capita is the strongest predictor. We note a distinct breakpoint in income level at around USD 10,000 per capita (equivalent to Mexico and Turkey in 2018), beyond which the prediction declines markedly, suggesting an inverse relationship between development and incentives for fleeing, all else equal. However, the size of the national economy—along with education level and infant mortality rate—usually changes little between years, whereas the more volatile inter-annual economic growth indicator exhibits a much weaker effect. Taken together, this suggests that the measured socioeconomic conditions shape latent migration potential but contribute less to temporal fluctuations in asylum migration to the EU.

The ALE plots reveal consistent but weak marginal predictive power for the climate indicators (Fig. 4d). Temperature is mostly negatively associated with asylum migration but the range of observed values for individual countries is small, entailing a modest substantive effect on the out-of-sample prediction. This is in some contrast to the variable's statistically significant effect in regression models (Supplementary Table 2) and the in-sample performance on the RF model's recall (Supplementary Fig. 2), where the small temporal changes of temperature, compared to its cross-sectional variation, makes it better able to separate

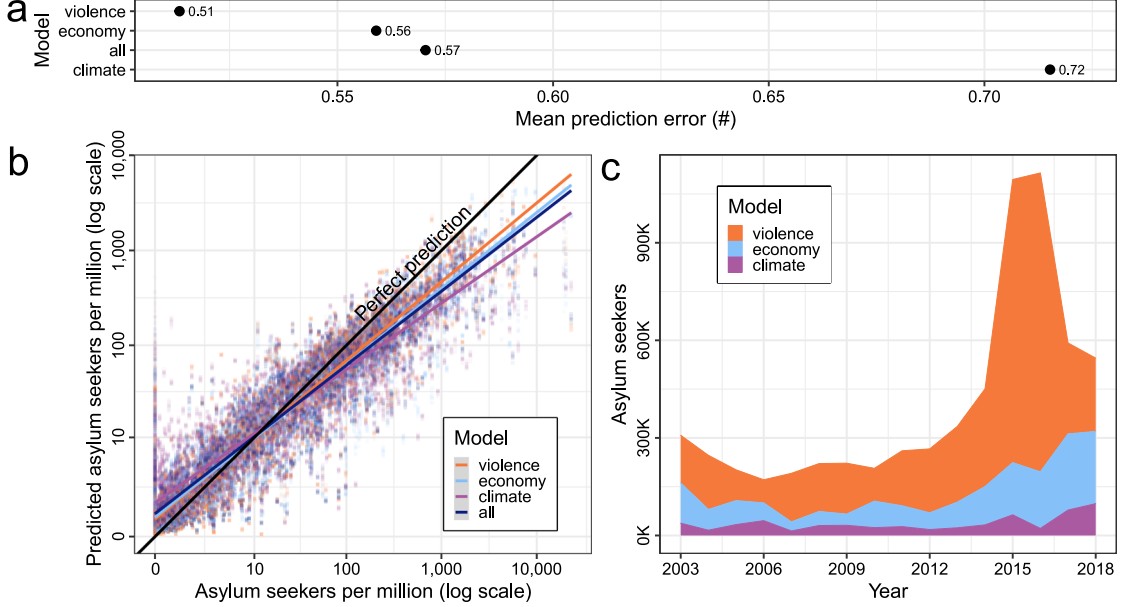

**Fig. 3 Out-of-sample prediction performance by model. a** Mean out-of-sample prediction error of random forest (RF) component models. **b** Predicted vs. observed numbers of asylum applications with imposed model average linear effects for the violence component model (orange), economy model (light blue), climate model (purple), and the complete model (dark blue). **c** Share of annual global volume of asylum seekers best predicted by each RF component model. The results in **a**–**c** are generated from leave-future-out cross-validation, trained on alternative 4-year subsets of empirical data for the period 1999–2017, and tested against observed outcomes for the subsequent year 2003–2018. $N = 3413$ country-year observations, examined over 160 simulations.

between low and high producers of asylum migrants. In that sense, the results from the LFO-CV analysis serve as a useful reminder that seemingly important regressors may be poor predictors on new data[14–16]. The shapes of the ALEs for the dynamic SPEI indicators, which measure inter-annual and smoothed multi-year drought over agriculture-producing areas, are virtually flat. This implies that there is little to gain from monitoring climate signals—either alone or in interaction with other important structural factors—if we seek to predict near-future increases in asylum seeker flows to the EU. Although this does not rule out the possibility that adverse climatic conditions influence asylum seeker flows in specific contexts[11], a powerful and reliable short-term climate effect would have been picked up by the model even if it had worked indirectly via, e.g., increased violence or depressed agricultural incomes. Separately, the evidence for a general climate-conflict link, which could trigger significant forced displacement, is judged to be modest[49–51].

In the Supplementary Information, we document a range of robustness tests. First, we estimate the models without the baseline indicators to better judge the difference in predictive power between the theoretical components. As shown in Supplementary Section 3.1, all models lose accuracy when the contextual factors are dropped and the gap between the violence and economy models and the inferior climate model also increases. In Supplementary Sections 3.2–3.3, we consider alternative lag structures for the predictors to account for the fact that the time from a shock occurs and a decision to leave is made until the migrant arrives in Europe varies considerably. The results for contemporaneous and 2-year lagged effects are very similar to those presented in Fig. 3. In subsequent tests, we alter the duration of the sliding training periods (up to 12 years) and the temporal gap between the training and test samples (up to 8 years into the future) (see Supplementary Sections 3.4–3.7). Further, we re-run all prediction models on absolute numbers of asylum applications per capita instead of log-transformed values (Supplementary Section 3.8) and we estimate global prediction

models using yearly stocks of asylum applications (Supplementary Section 3.9). Lastly, we estimate conventional in-sample regression models (Supplementary Section 3.10). Collectively, these tests bolster the results presented here and demonstrate that the comparatively weak predictive performance of climatic conditions on asylum migration is not a result of particular modeling decisions.

## Discussion
Conditions in countries of origin, although important, do not provide a complete view of the determinants of asylum migration. The availability of safe and sustainable havens in the immediate neighborhood lowers the incentives for long-distance mobility, and aspiring migrants' access to network, information, and resources also informs their decisions about whether, when, and where to go[20,52]. Moreover, non-quantifiable changes in immigration policies in major destination areas can have profound impacts on arrival statistics[21,53]. For example, the above-noted drop in asylum applications to EU-28 since 2016 (Fig. 1a) is not solely a response to declining conflict severity in the Middle East but also partly the result of an agreement between the EU and Turkey that abated the Aegean transit route[17]. Likewise, the closing of international borders in response to the COVID-19 pandemic reduced the number of new asylum applications submitted to the EU in 2020 by one third[54].

International forced migration, such as asylum migration, is neither the only nor the most plausible type of migratory response to adverse environmental conditions. Most migration in developing countries takes the form of internal rural–rural or rural–urban migration, and cross-border displacement typically is confined to the immediate neighborhood. Only a small subset of migrants is able to travel long distances and apply for asylum in Europe or the United States. Although a reasonable expectation is that climate-driven forced migration will reinforce rather than fundamentally alter these trends[10], data limitations presently hinder a systematic investigation into the relative contribution of

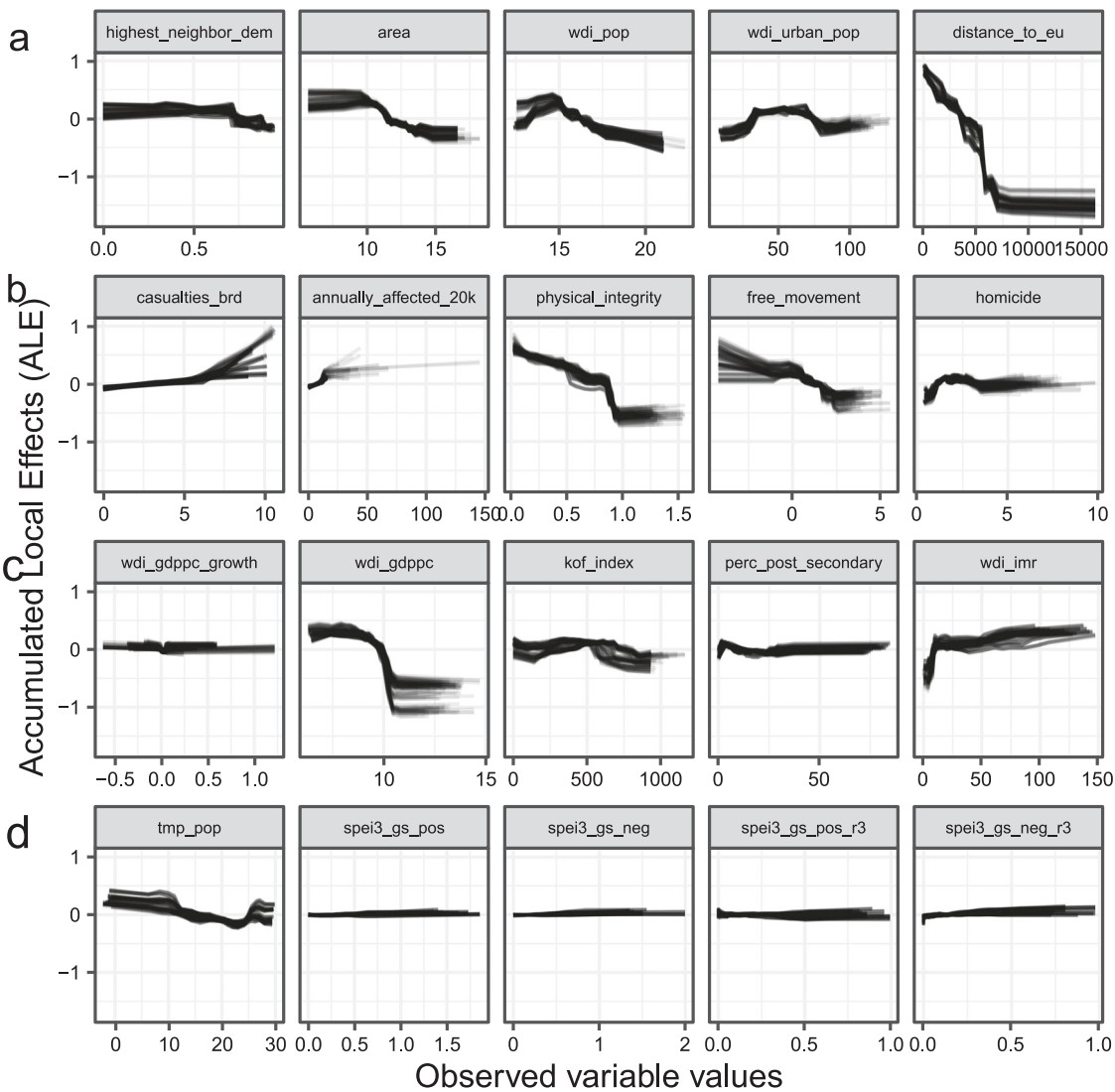

**Fig. 4 Accumulated Local Effects (ALEs) of asylum migration predictors. a** Baseline component indicators. **b** Violence component indicators. **c** Economy component indicators. **d** Climate component indicators. ALEs give the marginal difference in prediction with an incremental change in the predictor. *Y* axis values represent change in log asylum applications per capita. The results in **a**–**d** are generated from leave-future-out cross-validation, trained on alternative 4-year subsets of empirical data for the period 1999–2017, and tested against observed outcomes for the subsequent year 2003–2018. *N* = 3413 country-year observations, examined over 160 simulations.

climate and non-climatic determinants in predicting internal displacement in the manner conducted here[55]. Likewise, our results do not refute the possibility of long-term indirect effects of climate change or the well-founded concerns over disastrous consequences of climate-change impacts that might arise in later parts of the century[49]. A recent report by the Intergovernmental Panel on Climate Change concludes that climate change exceeding local adaptation limits can trigger escalating losses and result in forced migration, although the stated confidence is low, reflecting the lack of systematic research into the magnitude of such impacts[56]. This constitutes an important priority for future research.

Addressing the challenges of global migration by improving conditions in countries of origin is a fundamental pillar of the UN Global Compact on Refugees[57] and the UN Global Compact on Migration[58], but effective policy interventions require a rigorous understanding of the key drivers of migration. The results presented here reveal that explanations rooted in climatic conditions add little to our ability to predict asylum migration, despite their increasing prominence in contemporary academic and public

discourses[11,13,59]. Instead, we demonstrate that exposure to political violence and extensive restrictions on civil liberties constitute important, politically relevant early warning signals of future waves of forced cross-border migration. This is an important and positive insight, as it shows that asylum seekers are continuing to use the system as intended, even though the majority of applicants to the EU ultimately are denied protection status[17,18]. In 2019, the overall rejection rate on first instance decisions was 61.9%, although with significant variation across EU members[60]. Around half of the successful applicants were granted refugee status with the remainder being given subsidiary protection status or residence permit for humanitarian reasons.

Inclusive, well-functioning political institutions have a range of well-documented advantages, including more stable economic growth, increased investments in human capital, and more equitable distribution of wealth and privileges[61–63]. Curbing political violence also has immediate benefits in terms of improving food security, facilitating economic recovery, and diminishing citizens' need for international protection[22,29]. Results presented here suggest that investments in conflict resolution and peacebuilding,

and strengthening the quality of governance in fragile or repressive states constitute important entry points for addressing future risks related to asylum migration.

## Methods

**Dataset construction.** The dataset structure is country-year, where each independent state in the international system (175 countries)[64] is observed once every calendar year for all valid years, 1999–2018 ($N$ = 3413 country-years)[65]. To permit the construction of country-level indicators from more detailed georeferenced data, we rely on the CShapes dataset[66], which provides time-varying spatial data on the delineation of all country boundaries. The CShapes data are currently only updated through 2016, but as no major changes to the international system occurred during 2017–2018, we extended the country-year boundary shapefiles by 2 years to maximize utility of the asylum flow data.

**Asylum applications.** The outcome variable measures the number of asylum applications submitted to any EU member state (EU-28) from each country of origin in each calendar year, 1999–2018, collected by Eurostat and obtained via the statistical office of the UNHCR. Unlike global asylum statistics, the EU data contain information on time of first application, which allows us to measure volatile flows of new arrivals. The country-year asylum estimates are divided by the population size of the origin country[67] to make numbers comparable across cases, and transformed via natural logarithm to limit outlier bias. A small number of asylum seekers from microstates are omitted due to lack of country membership in the international system[64]. Asylum seekers classified by the original data as originating in the occupied Palestinian territories (OPT) are recoded with Israel as the country of origin, as OPT lacks formal recognition as an independent country (likewise, Palestine's struggle for independence is conventionally coded as a civil conflict in the state of Israel). Western Sahara is treated as part of Morocco, whereas asylum seekers with registered origin in Tibet, Hong Kong, and Macao are recoded as originating from the People's Republic of China. These coding decisions do not imply an expression of opinion concerning the legal status of these territories or their authorities.

**Climate predictors.** The climate component consists of five country-year indicators that capture different dimensions of climate variability. We include population-weighted average yearly temperature (Celsius) in each country-year, "tmp_pop," generated by summarizing the product of a normalized population density raster from the Gridded Populations of the World (GPW) v.4 dataset[68], and gridded land surface air-temperature data from CRU TS 4.03[69]. To capture climate anomalies relevant to the agricultural sector, we rely on four complementary measures based on SPEI-3, calculated from the CRU data using the SPEI package in R. When aggregating SPEI scores to the country-year, each $0.5 \times 0.5$ decimal degree grid cell month is weighted by the share of the area within the cell that is in growing season, relative to the total area of the cell that has a specific growing season during a year, based on data from the MIRCA2000 dataset[70]. There are 12 weights for each cell, 1 per month, and the weights sum to 1 within each cell. Cells without cropland are ignored when aggregating to the country-year. From this, we create two country-year anomaly indicators: "spei3_gs_neg" reflects negative deviation (dryness) from normal meteorological conditions (SPEI-3 scores for wetter than normal conditions are recoded zero) and "spei3_gs_pos" reflects positive deviation (wetness; drier than normal recoded zero). To capture consecutive years of growing-season climate anomalies, we further include 3-year moving average SPEI scores, similarly separated between anomalously dry ("spei3_gs_neg_r3") and wet ("spei3_gs_pos_r3") periods. Supplementary Table 1 provides descriptive statistics of all indicators.

**Economy predictors.** The economy component contains five indicators that measure countries' socioeconomic performance and structure. To proxy economic level of development, we include log-transformed GDP per capita, "wdi_gdppc", expressed in constant 2011 purchasing power parity-adjusted International Dollars, derived from the World Bank's World Development Indicators (WDI) (NY.GDP.PCAP.PP.KD)[67]. From this, we calculate year-on-year growth in GDP per capita "wdi_gdp_growth". Extent of interpersonal globalization, "kof_index", is based on statistics on international voice traffic, interpersonal financial transfers, international tourism, student exchanges, and share of foreign-born residents. The sources of these data are ETH Zurich's KOF globalization database (KOFIpGIdf)[71]. Education levels, "perc_post_secondary", are measured as the percentage of the population aged 20–29 years with post-secondary or higher education, taken from the Wittgenstein Centre for Demography and Global Human Capital[72]. In addition, we include an indicator of infant mortality rate per 1000 live births, "wdi_imr", derived from WDI (SP.DYN.IMRT.IN)[67].

**Violence predictors.** The violence component consists of five predictors that capture complementary dimensions of conflict and persecution. Severity of armed conflict, "casualties_brd", is given as the log-transformed best estimate of the number of battle-related deaths from intrastate armed conflict in each country-year, derived from the UCDP/PRIO Armed Conflict Dataset v.19.1[47]. We also

estimate the log-transformed number of civilians exposed to deadly conflict, "annually_affected_20k", by drawing 20 km buffers around each conflict event recorded in the UCDP Georeferenced Event Dataset (GED) v.19.1[73] and summarizing the underlying population from the GPW raster, avoiding double-counting. As GED v.19.1 lacks the battle event data for Syria, we make use of a beta release of UCDP GED for the period since January 2016, whereas missing population exposure for 2011–2015 was replaced via multiple imputation. To account for violence below the level of civil war events, we include an indicator of physical integrity rights, "physical_integrity", measured as the extent to which citizens are safe from political killings and torture by government agents, based on the V-Dem v.9 dataset (v2x_clphy)[74]. Another proxy for political repression, "free_movement", measures the extent to which citizens are free to travel in and out of their country without being subject to restrictions or persecution by public authorities, obtained from V-Dem v.9 (v2clfmove)[74]. Finally, "homicide" measures the log-transformed number of people killed as a result of interpersonal violence, provided by the Global Burden of Disease 2017 project[75].

**Baseline indicators.** All models contain a set of baseline indicators that capture inert demographic and contextual factors that shape latent opportunities for fleeing the country of origin and filing an application for asylum in the EU. Extent of democratic neighborhood, "highest_neighbor_dem", gives the highest democracy score among land-contiguous neighbors and countries separated by a maximum of 400 nautical miles of water from the country of observation. Data on neighborhood contiguity were extracted from the COW Direct Contiguity Data v.3.2, whereas we use an index of electoral democracy (v2x_polyarchy) from V-Dem v.9 to measure regime characteristics among neighboring countries[74]. Log-transformed country area in km$^2$ ("area") and population size ("wdi_pop"), obtained from WDI[67], proxy possible substitution effects between internal and international migration. Extent of urbanization, measured as the percentage of the population that resides in urban settlements, "wdi_urban_pop", is also from WDI[67]. Finally, we include the minimum geodesic distance (km) from each country of observation to the nearest EU-28 member state, calculated using the CShapes country boundary data, in "distance_to_EU".

**Missingness and time lag.** Several indicators lack complete information for all country-years. To remedy missing data problems, we create ten alternative datasets where all missing values are replaced with imputed values via Amelia II[76]. All models are then estimated on all ten datasets, such that the variance in reported outcomes partly reflects uncertainty about missing values. The general sensitivity of the predictions to the imputation procedure is small. To ensure that the conditions in source countries are measured prior to the recording of asylum applications and to allow for time spent in transit, all predictors are imposed a 1-year time lag in the main specification.

**Model estimation and benchmarking.** To mimic the real-world challenge of predicting near-future asylum migration, we rely on LFO-CV with a RF machine-learning algorithm. This entails that all models are trained on subsets of the data and tested by predicting outcomes on future observations in the dataset. Preliminary tests revealed that models using 4-year sliding periods of training data to predict the subsequent year of asylum flows generated the most consistent results with efficient use of available data. For each subsample, decision trees are generated using subsets of available variables for each split. The leaf nodes of these decision trees hold expected values for a small number of corresponding observations. The conditionality of the splits ensures that both nonlinear functional forms and interactions between variables inform the prediction. In the complete 1999–2018 sample, we train the models on 16 partly overlapping samples (1999–2002, 2000–03, 2001–04, …, 2014–17) and predict outcomes for the following year within the range 2003–2018, repeating all simulations 10 times on slightly different datasets to reflect alternative imputations of missing values (total number of simulations for main specification: 160).

As an aggregate performance metric of the component models, we estimate mean absolute deviations between observed and predicted values for all models across all subsamples and alternative imputations. Better-performing models have lower average error scores. As prediction errors for individual observations are a function of the true numbers of asylum applications, this metric rewards models that perform well in predicting large-scale migration (expressed in logs of migrants per capita). Model results are aggregated over all observations (Fig. 3a), assessed as a function of observed values (Fig. 3b), and evaluated over time (Fig. 3c). In addition, we inspect the predictive contribution of each component indicator through ALEs (Fig. 4), which reveals the marginal influence over incremental value changes and consistency across subsamples.

**Reporting summary.** Further information on research design is available in the Nature Research Reporting Summary linked to this article.

## Data availability
All underlying data are freely available from the cited sources. The CShapes GIS data of country boundaries are available from https://cran.r-project.org/web/packages/cshapes/

index.html. UNHCR statistics on asylum applications by country of origin are available from https://www.unhcr.org/refugee-statistics/download/?url=E1ZxP4. SEDAC's Gridded Population of the World v.4 population raster data are available from https://sedac.ciesin.columbia.edu/data/collection/gpw-v4. Gridded monthly climate data (v.4) from the University of East Anglia's Climate Research Unit are available from https://crudata.uea.ac.uk/cru/data/hrg/. MIRCA2000 spatial data on crop areas and growing seasons are available from https://www.uni-frankfurt.de/45218023/MIRCA. Country-year statistics on Gross Domestic Product per capita, infant mortality rate, population size, land area, and urbanization are available from the World Bank's World Development Indicators: https://databank.worldbank.org/source/world-development-indicators. Data on interpersonal globalization are available from ETH Zurich's KOF Globalization Index: https://kof.ethz.ch/en/forecasts-and-indicators/indicators/kof-globalisation-index.html. The Wittgenstein Centre for Demography and Global Human Capital provides country-year data on education levels: http://dataexplorer.wittgensteincentre.org/wcde-v2/. Data on armed conflict (v.19.1) are available from the Uppsala Conflict Data Program: https://ucdp.uu.se/downloads/. Country-year data on individual liberties and electoral democracy (v.9) are available from the V-Dem Institute: https://www.v-dem.net/en/data/archive/previous-data/data-version-9/. The Global Burden of Disease Study 2017 provides statistics on homicide rates: http://ghdx.healthdata.org/gbd-2017. Data on interstate contiguity (v.3.2) are available from the Correlates of War Direct Contiguity Dataset: https://correlatesofwar.org/data-sets/direct-contiguity. Replication data to reproduce results reported here, which were created from these sources, are available from Harvard Dataverse: https://doi.org/10.7910/DVN/6WRMCO.

## Code availability

The analysis was conducted using the R statistical software system. Replication code is available from GitHub at https://github.com/prio-data/climsec_nc21_repcode.

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

## Acknowledgements
We thank W. Neil Adger, Christiane Fröhlich, Maria Franco Gavonel, François Gemenne, Elisabeth Gilmore, Tim Hatton, Vally Koubi, Idean Salehyan, Carlos Vargas-Silva, Nina von Uexkull, and participants at research seminars at PRIO, NTNU, and Uppsala University for comments on earlier drafts. This work was supported by the European Research Council through Consolidator Grant number 648291 awarded to H.B.

## Author contributions
S.S., J.V. and H.B. designed the study. S.S. and J.V. assembled the data and conducted the statistical analysis. S.S. prepared the initial draft of the manuscript. J.V. prepared data code and documentation. J.C. supervised migration data selection and contextualization. H.B. provided funding, coordinated the analysis and review process, and wrote the final manuscript. All authors contributed to the interpretation of the results and drafting of the manuscript.

## Competing interests
The authors declare no competing interests.
