## [Peer Review File · Nature Communications]

REVIEWER COMMENTS

Reviewer #1 (Remarks to the Author):

This was an interesting and well-written paper. I was particularly impressed with the care taken with regard to the statistical modelling. There are three main issues I would like addressed.

First, there is a need for a more balanced discussion on the effects of climate. I disagree that there is 'high scientific agreement' that climate variability increases migration. The references cited to back this statement come mainly from the environmental literature but I believe the social and economic literature would provide a different picture -- they would say that climate could be important but secondary to economic, social and political conditions. And there are lots of places that have extreme weather events that people move to and/or live in (e.g., Dubai, Las Vegas). Further, developed economies with resources and insurance can mitigate some of the climatic changes through building better homes, sea walls or air conditioning.

Second, in relation to the topic of asylum migration and climate, I found the study to be a very specific case. Asylum migration to Europe is the exception, not the norm. As the authors explain in the discussion section (p. 8 lines 25-30), most of the movements are short distance or cross-border. Areas sending migrants that are affected (and most likely to be affected) by climate change are far away from Europe Union countries. In other words, the results are not surprising as (1) it excludes most of the people directly affected and (2) climate reasons for migrating are not normally a valid basis for refugee status (in the current state of affairs).

Third, I liked how you disaggregated the relative model contributions in Figure C but wondered whether you considered including interactions between the violence, economy and climate variables? As stated in the paper, these things are not independent and you should be able to test for them using the model.

Minor comment: Figure 1B (map) is not very clear. Perhaps you could just show countries that sent flows over a certain threshold, e.g., 300K?

Reviewer #2

I enjoyed reading the paper, which aims to disentangle the relative importance of different types of drivers (climate component, economy component, and violence component) on asylum applications. The paper applies a machine learning prediction framework to overcome the difficulties that regression analyses have encountered in achieving this specific objective. Even if I find the objective of the paper very interesting, I have some concerns, which are discussed below.

1. I think that the authors did not do a good job in placing their contribution within the existing literature and more importantly in describing what is the motivation for conducting such analyses. At present the authors better describe the motivation for using predictive modelling approaches in the

Supplementary Information, than in the text. The advantage of applying predictive modelling compared to model-based associations in the context of climate-induced migration does not depend on the presence of collinearity or existence of rival theories, as currently stated in the main text. There are many micro-founded regression analyses of non-climatic and climatic drivers of migration and there are no conflicting issues between them.

I do think however that predictive modelling approaches can largely contribute to this literature because, as stated in the SI, several factors influence migration outcomes through complex indirect and conditional pathways, and “seeking to isolate, quantify, and rank their individual causal effects through in-sample regression analysis is probably unfeasible”. The picture below is a clear example of how the different drivers interact in shaping migration: environmental drivers influence migration directly, but also indirectly, by affecting social, economic demographic and political drivers. This specific feature gives rise to an over-controlling bias if one tries to assess the causal effects of all these drivers through regression analyses. By estimating reduced-form relationship between migration outcome and climatic variables only (Missirian and Schlenker, 2017; Beine and Parsons, 2017; Cattaneo and Peri, 2016), researchers avoid over-controlling bias, but cannot say something on the relative importance of the different drivers. I admit that I do not know predictive modelling approaches, and I trust the authors when they say that they can evaluate the relative influence of both climatic conditions and other conditions in predicting contemporary asylum migration, without incurring in a double-counting effect. The relationship of interest in fact is of the following form:

$Y=f(C, X(C))$ where Y = migration outcome, C is a matrix of climatic variables and X is a matrix of non-climatic variables, which are an outcome of C (Dell et al. 2014)

I think that the authors should revise the introduction as this point is not clear at all.

Source: Black, R., Bennett, S., Thomas, S. et al. (2011) "Migration as adaptation" Nature

Beine M. and C. R. Parsons (2017) "Climatic Factors as Determinants of International Migration: Redux", CESifo Economic Studies

Cattaneo C. and G. Peri (2016) "The migration response to increasing temperatures" Journal of Development Economics

Dell M., B. Jones and B. Olken (2014) "What Do We Learn from the Weather? The New Climate–Economy Literature", Journal of Economic Literature

2. Given the main objective of the paper, which is an evaluation of the relative importance of the different drivers – and not an evaluation of the causal effect of the possible drivers – I wonder why the authors decided to analyse asylum applications, rather than conventional flows of migrants. These two flows differ remarkably, in the areas where they are originated, but also in the motivations that drive the voluntary and non-voluntary decision to move. While the analyses of the causal effect of climatic drivers on asylum flows make sense, and thus the use of regression analyses (Missirian and Schlenker, 2017), I have some doubts that this specific outcome variable is the best in the current analysis, given the motivation of the paper. It seems quite obvious to me that political violence and repression are the most powerful predictors of asylum applications. A different picture might emerge if the authors consider conventional migration flows for their analysis. The authors refer to data limitation as a hindering factor for this extension, but I have reasons to doubt that this is a real limitation, in particular if one wants to study migration to OECD countries only. One good candidate is the OECD International Migration Database (Adsera, 2015). As a matter of fact, I would guess that 90% of regression analyses on migration drivers apply conventional flows of migrants.

Adsera, A. and M. Pytlikova (2015). The role of language in shaping international migration. The Economic Journal 125(09), 49–82

3. I would better describe the methodology: the authors list three sets of components (each comprising a set of variables). Are these separate components of the different specifications (namely, the climate model only contains the set of variables of the climate component plus the baseline indicators?) or do the authors build the models adding one component to another? For example, I do not understand the comment to Figure 3a: "The climate model is comparatively poor; adding the five climate indicators to the violence model increases the model's average prediction error". What would it be the model's

average prediction error if only the climate component is used, without the indicators that comprise the violence component?

4. It is not clear how the authors chose the indicators to be included in the climate component. Could the poor performance of the climate model be due to the inaccurate choice of (some of) the climatic indicators? First, while temperature should not raise concerns, SPEI might not be the most powerful predictor of migration. It could be that the inclusion of this specific indicator within the climate component drives down the overall influence of the climate component. Moreover, while negative SPEI is a good proxy for drought, it is not clear to me the role of positive SPEI. If the authors want to measure floods, for example, there are better indicators to be used (for example, measures taken from the top percentile of the rain distribution). What about the choice of the time scale for the SPEI? 3 months is probably too short as a three-month scale should detect meteorological droughts, while longer scale (between 6 and 12) are better used to detect agricultural droughts – which is the channel for migration. What about the use of population metrics to weight the temperature variable? While population-weighted temperatures are suitable to study conventional migration, they are less so to study asylum flows, which are often not drawn homogeneously from the country population. Some ethnic groups may be more prone to persecution than others, or unrest may occur in some specific location within the country.
5. How would the author reconcile the result of Figure S2, where temperature seems to score very well, and the results in Figure 3, where the whole climatic component produce the least accurate prediction? How can the authors discard the influence of temperature and conclude that temperature anomalies are weak predictors of asylum migration, given the evidence of Figure S2?

Minor

P.2, line 9: as already stated, I do not think that statistics are more complete for asylum application than for conventional migrants. This is not a sufficient motivation for focusing on asylum flows.

P.2, line 19. I do not agree with the statement. First, it is true that we lack understanding of the marginal effect of climatic drivers, RELATIVE to other determinants, but the present study does not fill this gap either, given that the results should not be interpreted causally. This sentence gives rise to expectations that are not filled out. Second, I do not think that this knowledge gap contributed to a rise in populism. Others are the drivers of populisms (Hainmueller et al, 2014).

Hainmueller, Jens, and Daniel J. Hopkins. 2014. "Public Attitudes Toward Immigration." *Annual Review of Political Science* 17 (1): 225–49

P.3, line 32: I think the authors quote the position of one single person - Andrej Mahecic- and not the UNHCR in general. The possibility to establish a climate-specific legal status, as it is for refugees, is highly debated within the UNHCR, due to the difficulties in accounting for the direct effect of climate drivers on migrations.

P. 3, line 10: please give figure of the rejection rate.

Reviewer #3 (Remarks to the Author):

Referee report for Climatic Conditions are weak predictors of asylum migration

Summary: The paper studies the determinants of asylum migration using random forests and leave-future-out cross-validation to determine the importance of climatic, economic and violence indicators. They find, unlike in recent literature, that climatic conditions are not good predictors of asylum migration while violence is the best predictor of asylum applications. The analysis is robust to a battery of robustness tests.

Comments: I enjoyed reading this paper. The question is very relevant. The authors are very good at explaining their motivation, methods and results.

1. Comments on the methods.

The authors say causal inference is necessary, but it is unsuitable for prediction. This statement has some truth in it, but I guess it depends what's the goal of those predictions. If the goal is policy design, the projections need to inform and be informed by a causal model. I agree with the authors that for a descriptive view of the world, we do not need causality methods but an excellent prediction model. Policy prescriptions have an implied causal relation. For example, the authors say, "improving political institutions should be a central element in society-wide climate adaptation in vulnerable regions." I added the emphasis. There is nothing in the paper that can support this policy proposal, even though it is a good policy idea. The last paragraph in the discussion is a policy prescription that cannot be derived from the current analysis.

The need for a causal model is also required to interpret the results in the paper. I am not implying that causal methods are needed, but we need to understand how the outcome variable relates to the explanatory variables and how those explanatory variables are associated with each other. If the authors want to conclude there isn't a relation between climate and migration, we need to know how climate affects the economy and conflict. There is a proven relationship between climate and conflict (Burke, M., S.M. Hsiang, E. Miguel. (2015). "Climate and Conflict", *Annual Review of Economics*. DOI: 10.1146/annurev-economics-080614-115430.) and a proven relation between conflict and migration (citations within the manuscript). Thus, there is a role for the climate to affect migration through conflict. The empirical model chosen by the authors, assumes the effects across climate, economy and violence indicators can be non-linear but must be contemporaneous (not with outcomes, but among themselves). The authors lag all the independent variables because there are months between the cause of migration and the resulting asylum application. Yet, this lag doesn't capture the possibility that there is an extemporaneous relation between climate and conflict that wouldn't show up as affecting lagged asylum applications.

I also have two concerns about aggregation. The first one is about geographic aggregation at the national level. I understand this is the level of aggregation required given the data available, so I am not asking to change data. I wonder how to present the results knowing that country-level aggregation masks a lot of the effect as most migration happens to close places and from rural to urban environments. The authors mention this, but they do not explain how their results could be affected by this reality and how valid their conclusion that climate doesn't affect asylum is. The second one is about time aggregation. There is a lot of variation across the year. There can be "partial-year" effects here where the results of drought in December 2010 affect outcomes in 2011 with the same assumed strength as events in January 2010 (Bernard, Andrew B., Esther Ann Boler, Renzo Massari, Jose-Daniel Reyes, and Daria Taglioni. 2017. "Exporter Dynamics and Partial-Year Effects." *American Economic Review*, 107 (10): 3211-28.). How does a drought in December 2009 show up as affecting asylum applications in 2010 or 2011?

2. Comments on the focus.

I know this is not my paper, but let me offer a candid take on the paper's focus. The paper's focus on

climate has me confused. I understand this is a response to the current research showing climate is a driver of asylum applications. Yet, the current version of the paper is not doing that. If this paper wants to be that critique of the literature, it needs to confront the methods and assumptions of other papers directly (citations 9, 10 and 47 in the manuscript).

Alternatively, the paper could focus on the actual predictors of asylum and show why these predictors are essential levers for policy design. The current version of the article doesn't address these issues either. For example, there are interesting results hidden in Fig 4, but there is no intuition given to explain why we should expect them. For example, why such a sharp drop in ALE for GDP per capita at around 10? Why the sharp decline in ALE for physical integrity around 1? Are these mechanical, what is the reason behind this? The paper is sitting between a critique and an independent contribution, and it is not delivering on either. The authors need to give the article one voice.

Let me close by reiterating that I think this is a good paper. I enjoy reading it and I hope my comments can help the authors strengthen paper.

REVIEWER 1:

“This was an interesting and well-written paper. I was particularly impressed with the care taken with regard to the statistical modelling.”

Response: Thank you for the encouraging overall assessment.

R1.1. “There are three main issues I would like addressed. First, there is a need for a more balanced discussion on the effects of climate. I disagree that there is 'high scientific agreement' that climate variability increases migration. The references cited to back this statement come mainly from the environmental literature but I believe the social and economic literature would provide a different picture -- they would say that climate could be important but secondary to economic, social and political conditions. And there are lots of places that have extreme weather events that people move to and/or live in (e.g., Dubai, Las Vegas). Further, developed economies with resources and insurance can mitigate some of the climatic changes through building better homes, sea walls or air conditioning.”

Response: Thank you for this feedback. The opening sentence of the abstract, which this comment cites, was not meant to signal that climate variability and change are considered the most important driver of contemporary migration, but we understand that it could be read that way. In the revision, this sentence now reads: “Recent research suggests that climate variability and change significantly affect forced migration, within and across borders.” This statement is backed up by an expanding volume of research, published in a range of disciplinary and general science journals, that finds environmental conditions to influence human mobility (although indeed there is less agreement about the strength of the climate effect)¹. Later in the introduction, where we describe how migration is generally understood as a result of complex interactions between a range of factors, we now clarify that climatic conditions often are considered secondary to socioeconomic and political ones. Hopefully, the manuscript comes across as more balanced now.

R1.2. “Second, in relation to the topic of asylum migration and climate, I found the study to be a very specific case. Asylum migration to Europe is the exception, not the norm. As the authors explain in the discussion section (p. 8 lines 25-30), most of the movements are short distance or cross-border. Areas sending migrants that are affected (and most likely to be affected) by climate change are far away from European Union countries. In other words, the results are not surprising as (1) it excludes most of the people directly affected and (2) climate reasons for migrating are not normally a valid basis for refugee status (in the current state of affairs).”

Response: It is true that asylum migration to Europe is the exception, not the norm. This is now highlighted in opening paragraph of the introduction (p. 2). We also agree that short-distance forced migration within the Global South deserves to be studied in its own right. It is entirely possible that the relative predictive importance of climatic conditions is higher for that form of migration, as alluded to by the reviewer. An intriguing hypothesis for future research! Yet, the prominence of the recent European ‘migrant crisis’, in combination with bold projections about climate change impacts on future asylum migration to the EU² and limited understanding of the relative importance of the climate effect, constitute compelling motives for studying predictors of asylum migration.

The number of migrants originating from Africa, the Middle East, and South Asia that have applied for asylum in the EU has risen sharply over the past decade (Fig. 1 in manuscript). Events in some of these sending areas, notably the Syrian civil war, have been associated with climatic extremes^{3,4}. Recently, adverse climatic conditions have been causally linked with asylum migration also at a more general level^{2,5}, and the notion of ‘climate refugees’ has gained considerable traction in media and policy circles alike⁶⁻⁸. Accordingly, we believe there are valid reasons to expect drought and temperature extremes to provide early warning signals of growth in the arrival of asylum seekers. See also our response to R2.2.

R1.3. “Third, I liked how you disaggregated the relative model contributions in Figure C but wondered whether you considered including interactions between the violence, economy and climate variables? As stated in the paper, these things are not independent and you should be able to test for them using the model.”

Response: Thank you for pointing to an aspect of our study that was unclear. Earlier research suggests that environmental impacts on migration are context dependent, so accounting for possible interactions with non-climatic factors is important. The Random Forest model that we rely on for the LFO-CV analysis by design explores all possible interactions between selected variables through an ensemble of decision trees and quantifies the total (direct and interactive) contribution of each variable to the model’s prediction. The estimated contribution of the climate component in Fig. 3 thus reflects both direct and interactive effects. Interactions between the components are accounted for in the ‘all’ model (which performs slightly worse on average than the violence model, Fig. 3A). We realize that this was not sufficiently clearly communicated in the original submission and have revised the text accordingly (see p. 4, as well as the new Methods section).

R1.4. “Minor comment: Figure 1B (map) is not very clear. Perhaps you could just show countries that sent flows over a certain threshold, e.g., 300K?”

Response: Thanks. Following the reviewer’s suggestions, we have replaced the continuous color scale in Fig. 1B with four discrete categories.

REVIEWER 2:

“I enjoyed reading the paper, which aims to disentangle the relative importance of different types of drivers (climate component, economy component, and violence component) on asylum applications. The paper applies a machine learning prediction framework to overcome the difficulties that regression analyses have encountered in achieving this specific objective. Even if I find the objective of the paper very interesting, I have some concerns, which are discussed below.”

Response: Thank you for the encouraging overall assessment.

R2.1. “I think that the authors did not do a good job in placing their contribution within the existing literature and more importantly in describing what is the motivation for conducting such analyses. At present the authors better describe the motivation for using predictive modelling approaches in the Supplementary Information, than in the text. The advantage of applying predictive modelling compared to model-based associations in the context of climate-induced migration does not depend on the presence of collinearity or existence of rival theories, as currently stated in the main text. There are many microfounded regression analyses of non-climatic and climatic drivers of migration and there are no conflicting issues between them. I do think however that predictive modelling approaches can largely contribute to this literature because, as stated in the SI, several factors influence migration outcomes through complex indirect and conditional pathways, and “seeking to isolate, quantify, and rank their individual causal effects through in-sample regression analysis is probably unfeasible”. The picture below is a clear example of how the different drivers interact in shaping migration: environmental drivers influence migration directly, but also indirectly, by affecting social, economic demographic and political drivers. This specific feature gives rise to an overcontrolling bias if one tries to assess the causal effects of all these drivers through regression analyses. By estimating reduced-form relationship between migration outcome and climatic variables only (Missirian and Schlenker, 2017; Beine and Parsons, 2017; Cattaneo and Peri, 2016), researchers avoid overcontrolling bias, but cannot say something on the relative importance of the different drivers. I admit that I do not know predictive modelling approaches, and I trust the authors when they say that they can evaluate the relative influence of both climatic conditions and other conditions in predicting contemporary asylum migration, without incurring in a double-counting effect. The relationship of interest in fact is of the following form:

$Y=f(C, X(C))$ where Y = migration outcome, C is a matrix of climatic variables and X is a matrix of non-climatic variables, which are an outcome of C (Dell et al. 2014)

I think that the authors should revise the introduction as this point is not clear at all.”

Response: This is a great comment that has helped us sharpen the presentation of central design decisions in the manuscript. The revision provides a more elaborate discussion of challenges with evaluating relative variable importance in complex, interactive systems through conventional regression analysis, why such models often perform poorly in predicting outcomes out of sample, and how the RF model overcomes these challenges (see, in particular, p. 4 plus Methods section). This innovation is crucial in order to assess the real-world relevance of climatic conditions in predicting near-future asylum migration to Europe. See also our response to R3.4.

R2.2. “Given the main objective of the paper, which is an evaluation of the relative importance of the different drivers – and not an evaluation of the causal effect of the possible drivers – I wonder why the authors decided to analyse asylum applications, rather than conventional flows of migrants. These two flows differ remarkably, in the areas where they are originated, but also in the motivations that drive the voluntary and non-voluntary decision to move. While the analyses of the causal effect of climatic drivers on asylum flows make sense, and thus the use of regression analyses (Missirian and Schlenker, 2017), I have some doubts that this specific outcome variable is the best in the current analysis, given the motivation of the paper. It seems quite obvious to me that political violence and repression are the most powerful predictors of asylum applications. A different picture might emerge if the authors consider conventional migration flows for their analysis. The authors refer to data limitation as a hindering factor for this extension, but I have reasons to doubt that this is a real limitation, in

particular if one wants to study migration to OECD countries only. One good candidate is the OECD International Migration Database (Adsera, 2015). As a matter of fact, I would guess that 90% of regression analyses on migration drivers apply conventional flows of migrants.”

Response: This comment, which overlaps with R1.2, helped us realize that the motivation of the study was insufficiently explained. To be clear, we are particularly interested climatic predictors of asylum migration to Europe, given the prominence of this ‘crisis’ in the media, its many social and political effects, and the fact that this particular migration flow has been causally linked to climatic conditions in countries of origin². We share the reviewer’s a priori expectation that political violence and repression are likely more powerful predictors of asylum applications, but this is yet to be demonstrated scientifically. Meanwhile, end-of-century extrapolations of climate-driven asylum migration under *ceteris paribus* have created alarmist headlines^{9,10}, and myths and controversies surrounding ‘climate refugees’ abound^{11,12}. We hope that our reasons for focusing on asylum migration is more clearly articulated in the revised introduction.

For our purpose, the statistics of first-time asylum applications to EU-28, collected by Eurostat and facilitated by UNHCR, represent the optimal data source. We agree that the relative influence of climate variables on other forms of migration also warrants scientific attention and are grateful to the reviewer for pointing to the inaccurate description of data limitations in our original manuscript. We have updated the text accordingly by referring specifically to lack of systematic data on internally displaced people (p. 12), which constitutes the form of human mobility most similar to the one considered here.

R2.3. “I would better describe the methodology: the authors list three sets of components (each comprising a set of variables). Are these separate components of the different specifications (namely, the climate model only contains the set of variables of the climate component plus the baseline indicators?) or do the authors build the models adding one component to another? For example, I do not understand the comment to Figure 3a: “The climate model is comparatively poor; adding the five climate indicators to the violence model increases the model’s average prediction error”. What would it be the model’s average prediction error if only the climate component is used, without the indicators that comprise the violence component?”

Response: Thank you. We have expanded on the description of the component models in the revised main text (p. 5–7), and the new Methods section offers additional details on measurements and model specifications.

To the reviewer’s question: We specify three distinct, theoretically informed components – climate (C), economy (E), violence (V) – each of which contains five indicators. In addition, we identify a set of baseline indicators (B), analogous to common controls in regression models, which define a country’s latent production of asylum migrants and which may condition the influence of the component indicators. The climate model thus consists of C + B, the economy model contains E + B, and the violence model is specified as V + B. For comparison, we also estimate a complete prediction model that makes use of all indicators. In Supplementary Information Section S3.1, we document results for models without B to better highlight differences in the components’ prediction performance when context is ignored. We

realize that the reviewer's confusion originated in our imprecise discussion of the results in relation to Fig. 3A; this has now been remedied.

R2.4. “It is not clear how the authors chose the indicators to be included in the climate component. Could the poor performance of the climate model be due to the inaccurate choice of (some of) the climatic indicators? [A] First, while temperature should not raise concerns, SPEI might not be the most powerful predictor of migration. It could be that the inclusion of this specific indicator within the climate component drives down the overall influence of the climate component. [B] Moreover, while negative SPEI is a good proxy for drought, it is not clear to me the role of positive SPEI. If the authors want to measure floods, for example, there are better indicators to be used (for example, measures taken from the top percentile of the rain distribution). [C] What about the choice of the time scale for the SPEI? 3 months is probably too short as a three-month scale should detect meteorological droughts, while longer scale (between 6 and 12) are better used to detect agricultural droughts – which is the channel for migration. [D] What about the use of population metrics to weight the temperature variable? While population-weighted temperatures are suitable to study conventional migration, they are less so to study asylum flows, which are often not drawn homogeneously from the country population. Some ethnic groups may be more prone to persecution than others, or unrest may occur in some specific location within the country.”

Response: We acknowledge that the original manuscript was not explicit enough in the theoretical rationale behind the specification of the components. Thanks to the extra space made available for the revision (E.4), the revised manuscript now provides more information to help the reader follow the presentation of the analysis and the results. To the four specific issues raised by the reviewer (marked in square brackets in the comment above):

[A] It is clear from the LFO-CV analysis that none of the SPEI-3 specifications contributes substantively to the climate model's predictive performance (Fig. 4 in manuscript). Models based on alternative lag structures and training/test sample specifications, documented in Supplementary Information, demonstrate that this result is not an artifact of arbitrary modeling decisions. Although the meagre effect of SPEI could be masking some of the influence of temperature in the climate model, random forest models seldom become worse by including additional variables.

Failing to detect a predictive signal of SPEI on future asylum applications is an important discovery in itself since theoretical and empirical literature suggests that drought is a relevant factor in human migration.^{13,14} Of course, it might be that drought is associated with asylum migration in more subtle ways than our growing-season-based predictors are able to capture, although the likelihood that such an effect is a relevant early warning predictor is small (recall that the RF model considers both non-linear functional forms and interactive effects). In response to this comment, we have expanded the discussion on the relative performance of the different component indicators in relation to Fig. 4 (p. 9–10); see also our response to R2.5.

[B] The second element in the reviewer's comment concerns the theorized role of positive SPEI values. Again, owing to the brief description of these indicators, this was not clear in the original submission. Positive SPEI is not included to capture floods and related rapid-onset hazards (for which it would be a poor proxy) but instead meant to represent beneficial conditions for agriculture. In rain-fed agricultural systems, above-average precipitation levels

during the growing season generally are associated with increased yields^{15,16}, which should reduce affected populations' incentive to migrate. This is now mentioned on p. 5–6. Floods frequently trigger displacement but are less commonly associated with migration¹⁴.

[C] Third, the reviewer asks about the temporal range for the measurement of drought. Since we are interested in capturing climatic shocks to agricultural economies and livelihoods, we rely on monthly estimates of the SPEI-3 index, measured during the growing season months of agricultural areas within each source country. By using the three-month variant, we account for climatic conditions shortly prior to and including the current growing season and ignore weather conditions during other parts of the year, which have less influence on the harvest in rainfed agricultural systems. To capture multi-year droughts, we also include three-year moving average SPEI-3 scores, measured to reflect the situation during consecutive growing seasons. This is now stated on p. 5–6.

In all, our study considers a more comprehensive set of drought indicators than previous migration studies. In addition to the yearly and three-year smoothed SPEI specifications, we test different lag structures (0–2 y.) and different temporal gaps between the training and test samples (up to 8 y.) to allow for further delays in the drought impact on asylum migration. These additional tests are documented in Supplementary Information.

[D] Lastly, the reviewer asks about the rationale behind the population-weighting of the temperature variable. The reason is simple: whereas SPEI-3 is tuned to capture agricultural shocks, and hence is measured exclusively for crop-producing areas and growing-season months within each country-year, climatic extremes also can affect non-agricultural economic activities, perhaps most prominently through adverse heat impact on worker productivity¹⁷. The best way to capture such effects is through population-weighting of the high-resolution temperature data. This is now mentioned on p. 6.

R2.5. “How would the author reconcile the result of Figure S2, where temperature seems to score very well, and the results in Figure 3, where the whole climatic component produce the least accurate prediction? How can the authors discard the influence of temperature and conclude that temperature anomalies are weak predictors of asylum migration, given the evidence of Figure S2?”

Response: Thanks, this is another excellent comment. It is true that temperature performs better than the SPEI-based indicators in predicting future asylum flows (see also our response to R2.4), and tests of in-sample variable importance suggests that temperature is the sixth most influential variable overall (Fig. S2). This fact was not properly acknowledged in the original submission but has been remedied (p. 10). Yet, Fig. 3 and 4 in the main manuscript and Fig. S2 in Supplementary Information show different qualities. The former reflect the components' and variables' contribution to the models' out-of-sample prediction, respectively, which is what we are primarily interested in. The latter reflects the variables' influence on in-sample recall. In in-sample recall, temperature plays an important function: The variable is accurately measured and differs considerably between countries while displaying much less variation over time within countries. It also correlates highly with variables from other components (see Fig. S1). As RF inductively subdivides the parameter space for included variables, temperature takes a prominent spot to identify countries with systematically different levels of asylum migrants. The analog in a GLM-type model would be a fixed effects

specification that estimates country-specific marginal effects. In other words, models end up relying on temperature for this structure when temperature is available, but similar relevant information can be deduced from other variables (e.g., physical integrity, GDP per capita) without loss of performance.

Evidently, the baseline in-sample effect of temperature does not translate into equally high out-of-sample predictive performance. The non-linear functional forms and statistical interactions fitted to any one training dataset help to recall asylum seeker numbers within that dataset, but they do not reliably predict future levels in the LFO-CV benchmarks. We cannot use this approach to judge whether temperature is a causal factor in asylum migration, because it correlates and interacts with other relatively stable variables over longer time periods. However, we can conclude that climatic factors are less relevant immediate factors because models predict just as well or better without having access to such variables.

In addition, Reviewer 2 provided four minor comments:

R2.6. “P.2, line 9: as already stated, I do not think that statistics are more complete for asylum application than for conventional migrants. This is not a sufficient motivation for focusing on asylum flows.”

Response: Thanks. We have clarified the motivation behind our focus on asylum migration in the revised introduction. We also have updated the discussion of data limitations (p. 12). That said, we contend that asylum statistics are likely to be more accurate and complete than data on most other forms of spontaneous human mobility (e.g., internal or cross-border displacement, irregular migration). Regulated forms of international migration (e.g., labor migration, family reunification, or education) are more directly shaped by policy, which means that discerning possible effects of climate variability presents a different form of challenge.

R2.7. “P.2, line 19. I do not agree with the statement. First, it is true that we lack understanding of the marginal effect of climatic drivers, RELATIVE to other determinants, but the present study does not fill this gap either, given that the results should not be interpreted causally. This sentence gives rise to expectations that are not filled out. Second, I do not think that this knowledge gap contributed to a rise in populism. Others are the drivers of populisms (Hainmueller et al, 2014).”

Response: The reviewer is correct that our study does not quantify the relative causal effect of climate vs. other factors in driving asylum migration. Rather, we seek to evaluate the relative predictive performance of climatic conditions, relative to other theorized drivers. To minimize confusion, we have removed the statement referred to by the reviewer and expanded the discussion on the distinction between causal and predictive modeling (notably, p. 5). Predictors that perform well on new data are likely to capture important data-generating processes underlying the theoretically informed causal variables whereas estimated causal effects that fail to predict similar outcomes out of sample might reflect overfitting or misspecification of the original model.

We also accept that our statement on the origins of rising populism in Europe was imprecise. Recent research points to the rapid growth in the arrival of asylum seekers as a driver of right-

wing populism¹⁹, not a lack of knowledge about leading drivers of this migration flow as the original sentence claimed. This has been corrected (p. 3).

R2.8. “P.3, line 32: I think the authors quote the position of one single person - Andrej Mahecic- and not the UNHCR in general. The possibility to establish a climate-specific legal status, as it is for refugees, is highly debated within the UNHCR, due to the difficulties in accounting for the direct effect of climate drivers on migrations.”

Response: The source is indeed a briefing note delivered by UNHCR spokesperson, with the note that media outlets can attribute any quoted text to him. However, the point that is conveyed in the note is presented as “a more detailed UNHCR assessment of this ruling” (i.e. the preceding ruling by the UN Human Rights Committee) and opens with the following sentence: “UNHCR has consistently stressed that people fleeing adverse effects of climate change and the impact of sudden and slow-onset disasters may have valid claims for refugee status under the 1951 Refugee Convention or regional refugee frameworks.” In other words, this is not an endorsement for the controversial establishment of a new legal category, but rather a call for a broader interpretation of existing frameworks. Our reference to the connection that the UNHCR makes between climate change and possibly valid claims for refugee status is pertinent and appropriately documented. However, there is clearly much more to say about positions within and beyond UNHCR, which lies beyond the scope of this paper.

R2.9. “P. 3, line 10: please give figure of the rejection rate.”

Response: We have added the latest rejection rate (2019) to the discussion (p. 12).

REVIEWER 3:

“I enjoyed reading this paper. The question is very relevant. The authors are very good at explaining their motivation, methods and results.”

Response: Thank you for the encouraging overall assessment.

R3.1. “The authors say causal inference is necessary, but it is unsuitable for prediction. This statement has some truth in it, but I guess it depends what's the goal of those predictions. If the goal is policy design, the projections need to inform and be informed by a causal model. I agree with the authors that for a descriptive view of the world, we do not need causality methods but an excellent prediction model. Policy prescriptions have an implied casual relation. For example, the authors say, "improving political institutions should be a central element in society-wide climate adaptation in vulnerable regions." I added the emphasis. There is nothing in the paper that can support this policy proposal, even though it is a good policy idea. The last paragraph in the discussion is a policy prescription that cannot be derived from the current analysis.”

Response: We realize that our discussion of causal analysis versus prediction led to some confusion. In the revised manuscript (p. 5), we elaborate on the motive and purpose of prediction and how such analysis can provide insights into the generalizability of causal effects reported in the empirical literature. Likewise, the concluding discussion now makes a clearer distinction between implications that can be derived directly from the prediction analysis and more overarching political issues that this research agenda speaks to.

R3.2. “The need for a causal model is also required to interpret the results in the paper. I am not implying that causal methods are needed, but we need to understand how the outcome variable relates to the explanatory variables and how those explanatory variables are associated with each other. If the authors want to conclude there isn't a relation between climate and migration, we need to know how climate affects the economy and conflict. There is a proven relationship between climate and conflict (Burke, M., S.M. Hsiang, E. Miguel. (2015). "Climate and Conflict", *Annual Review of Economics*. DOI: 10.1146/annurev-economics-080614-115430.) and a proven relation between conflict and migration (citations within the manuscript). Thus, there is a role for the climate to affect migration through conflict. The empirical model chosen by the authors, assumes the effects across climate, economy and violence indicators can be non-linear but must be contemporaneous (not with outcomes, but among themselves). The authors lag all the independent variables because there are months between the cause of migration and the resulting asylum application. Yet, this lag doesn't capture the possibility that there is an extemporaneous relation between climate and conflict that wouldn't show up as affecting lagged asylum applications.”

Response: Thanks, this comment raises important issues. To be clear, our analysis does not seek to demonstrate (or falsify) particular causal relationships but rather to evaluate an important predictive implication of extant empirical research, namely the importance of climatic variables in providing early warning signals of new asylum migration to the EU. Conventional regression analysis is unsuited to assess the relative influence of competing, collinear, and endogenous explanations on an outcome such as this²⁰. This is now explicitly mentioned on p. 4–5, and results from a conventional regression analysis are now briefly discussed on p. 7 as a point of departure for the prediction analysis. See also our responses to comments R2.1, R2.7, and R3.1.

The reviewer is correct in pointing out that our research is designed to detect short-term early warning signals of climatic, economic, and political shocks on migration to the EU. However, we note that we investigate both contemporaneous, one-year, and two-year lagged effects (as well as three-year moving average SPEI effects), and we further explore different temporal gaps between the training and test samples (up to eight years). Although all predictors within a model are assigned equal time lag, this is consistent with how the climate is assessed in the empirical literature, where significant effects have been reported (see also Table S2). Taken together, our prediction analysis provides a more comprehensive and flexible approach than comparable earlier research.

In response to this comment, we now state more clearly (p. 12) that our analysis does not refute the possibility of long-term predictive signals of climatic conditions on asylum applications, nor does it imply that climate change cannot contribute to increasing the flow of asylum migration to Europe in the future.

R3.3. “I also have two concerns about aggregation. The first one is about geographic aggregation at the national level. I understand this is the level of aggregation required given the data available, so I am not asking to change data. I wonder how to present the results knowing that country-level aggregation masks a lot of the effect as most migration happens to close places and from rural to urban environments. The authors mention this, but they do not explain how their results could be affected by this reality and how valid their conclusion that climate doesn't affect asylum is. The second one is about time aggregation. There is a lot of variation across the year. There can be "partial-year" effects here where the results of drought in December 2010 affect outcomes in 2011 with the same assumed strength as events in January 2010 (Bernard, Andrew B., Esther Ann Boler, Renzo Massari, Jose-Daniel Reyes, and Daria Taglioni. 2017. "Exporter Dynamics and Partial-Year Effects." *American Economic Review*, 107 (10): 3211-28.). How does a drought in December 2009 show up as affecting asylum applications in 2010 or 2011?”

Response: These are great comments that point to challenges with the conventional country-year approach too often ignored. To minimize adverse effects of spatial over-aggregation, we exploit high-resolution meteorological data, available at 0.5 x 0.5 decimal degrees at global scale, which are aggregated to the country level via theoretically informed weights (i.e., SPEI is measured exclusively for cropland to proxy shocks to rural livelihoods; temperature is weighted by local population density to capture adverse impacts on urban economies). Likewise, since weather sensitivity of agriculture varies across seasons, the SPEI indicators only capture anomalies during the local growing season months. We elaborate on these operationalization choices in the expanded Results discussion (p. 5–7) and in the Methods section.

In principle, we could have evaluated the predictive signal of climatic anomalies at a monthly level since monthly statistics of new asylum applications are provided by Eurostat. However, most economy and violence predictors are only measured by calendar-years. Besides, within-year variations in departure dates from countries of origin do not translate directly into similar variation in time of arrivals in Europe since several major transit routes exhibit distinct seasonal patterns. For instance, crossing of the Mediterranean is more viable during the summer months. Annual averages even out such artifacts and one-year lags help account for time in transit (p. 6–7).

R3.4. “I know this is not my paper, but let me offer a candid take on the paper's focus. The paper's focus on climate has me confused. I understand this is a response to the current research showing climate is a driver of asylum applications. Yet, the current version of the paper is not doing that. If this paper wants to be that critique of the literature, it needs to confront the methods and assumptions of other papers directly (citations 9, 10 and 47 in the manuscript). Alternatively, the paper could focus on the actual predictors of asylum and show why these predictors are essential levers for policy design. The current version of the article doesn't address these issues either. For example, there are interesting results hidden in Fig 4, but there is no intuition given to explain why we should expect them. For example, why such a sharp drop in ALE for GDP per capita at around 10? Why the sharp decline in ALE for physical integrity around 1? Are these mechanical, what is the reason behind this? The paper is sitting between a critique and an independent contribution, and it is not delivering on either. The authors need to give the article one voice.”

Response: Thank you for your opinion, which is greatly appreciated. Either of the two alternative routes would make for an interesting and timely study. What we seek to do is close to the first approach. Yet, we believe it is defensible – indeed, desirable – to cover the middle ground, too. Present research (e.g., Missirian and Schlenker; MS) estimates the causal effect of climate on contemporaneous asylum migration but does not offer insights into the real-world relevance of this effect, i.e., how important temperature is when compared to other drivers of forced migration. Our critique of their approach is not founded on a belief that they chose the ‘wrong’ model. Rather, we argue that fixed effects regression models are unsuited for out-of-sample prediction. Instead, we should research which variables have predictive capabilities beyond the sample and inform research and policy along these lines. This is now discussed in more detail as motivation for our predictive approach (p. 4–5).

Our finding does not undermine the causal interpretation of the result reported by MS, but it places their result in context. One important predictive implication of MS is that adverse climatic conditions in source countries trigger a near instantaneous growth in the migration of asylum seekers from that country to Europe. We find only weak indication that this is the case, implying that the MS result, while statistically significant in certain in-sample regression specifications, has modest influence on real-world fluctuations in the arrival of asylum seekers. See also our discussion in relation to R2.5.

Lastly, we agree with the reviewer that there is more to the story than we are able to convey here. The ALE plots, in particular, reveal many intriguing patterns that deserve further scrutiny, although space constraints mean that we have to defer such inquiries to future research.

R3.5. “Let me close by reiterating that I think this is a good paper. I enjoy reading it and I hope my comments can help the authors strengthen paper.”

Response: Thank you, we are confident that the comments from the reviewers have helped strengthen the paper.

REFERENCES

1. Hoffmann, R., Dimitrova, A., Muttarak, R., Crespo Cuaresma, J. & Peisker, J. A meta-analysis of country-level studies on environmental change and migration. *Nature Climate Change* **10**, 904–912 (2020).
2. Missirian, A. & Schlenker, W. Asylum applications respond to temperature fluctuations. *Science* **358**, 1610 (2017).
3. Gleick, P. H. Water, Drought, Climate Change, and Conflict in Syria. *Wea. Climate Soc.* **6**, 331–340 (2014).
4. Kelley, C. P., Mohtadi, S., Cane, M. A., Seager, R. & Kushnir, Y. Climate change in the Fertile Crescent and implications of the recent Syrian drought. *Proceedings of the National Academy of Sciences* **112**, 3241–3246 (2015).
5. Abel, G. J., Brottrager, M., Crespo Cuaresma, J. & Muttarak, R. Climate, conflict and forced migration. *Global Environmental Change* **54**, 239–249 (2019).
6. Reuters. World needs to prepare for ‘millions’ of climate displaced: U.N. (2020).
7. Deutsche Welle. Building walls to keep climate refugees out. (2019).
8. National Geographic. Future Warming Could Worsen Europe’s Refugee Crisis. (2017).
9. Devastating climate change could lead to 1m migrants a year entering EU by 2100. *The Guardian* (2017).
10. Climate change will drive one million refugees to Europe by 2100. *The New York Post* (2017).
11. Boas, I. *et al.* Climate migration myths. *Nat. Clim. Chang.* **9**, 901–903 (2019).
12. White, G. “Climate Refugees”—A Useful Concept? *Global Environmental Politics* **19**, 133–138 (2019).
13. Black, R. *et al.* The effect of environmental change on human migration. *Global Environmental Change* **21**, S3–S11 (2011).
14. Kaczan, D. J. & Orgill-Meyer, J. The impact of climate change on migration: a synthesis of recent empirical insights. *Climatic Change* **158**, 281–300 (2020).
15. Ray, D. K., Gerber, J. S., MacDonald, G. K. & West, P. C. Climate variation explains a third of global crop yield variability. *Nature Communications* **6**, 5989 (2015).
16. Fishman, R. More uneven distributions overturn benefits of higher precipitation for crop yields. *Environ. Res. Lett.* **11**, 024004 (2016).
17. Dell, M., Jones, B. F. & Olken, B. A. Temperature Shocks and Economic Growth: Evidence from the Last Half Century. *American Economic Journal: Macroeconomics* **4**, 66–95 (2012).
18. Acevedo Mejia, S. *The Effects of Weather Shocks on Economic Activity: What are the Channels of Impact?* (International Monetary Fund, 2018).
19. Brubaker, R. Why populism? *Theor Soc* **46**, 357–385 (2017).
20. Cranmer, S. J. & Desmarais, B. A. What Can We Learn from Predictive Modeling? *Political Analysis* **25**, 145–166 (2017).

REVIEWERS' COMMENTS

Reviewer #2 (Remarks to the Author):

The authors have addressed all my comments.

Reviewer #3 (Remarks to the Author):

Dear Authors,

Thank you for responding to my comments and queries. I think the paper is much improved in clarifying and presentation and I believe it will be a strong contribution to the literature.

Congratulations on a very well executed manuscript.